# Implications of the Harmonization of [18F]FDG-PET/CT Imaging for Response Assessment of Treatment in Radiotherapy Planning

Elisa Jiménez-Ortega [1,2] , Raquel Agüera [1], Ana Ureba [2,3] , Marcin Balcerzyk [1,4] , Amadeo Wals-Zurita [5], Francisco Javier García-Gómez [6] and Antonio Leal [1,2,*]

1 Departamento de Fisiología Médica y Biofísica, Universidad de Sevilla, 41009 Seville, Spain; elisajimenez@us.es (E.J.-O.); raguera@saludcastillayleon.es (R.A.); mbalcerzyk@us.es (M.B.)
2 Instituto de Biomedicina de Sevilla, IBiS, 41013 Seville, Spain; ana.ureba@ki.se
3 Medical Radiation Physics, Department of Physics, Stockholm University, 114 21 Stockholm, Sweden
4 Centro Nacional de Aceleradores (CNA), Universidad de Sevilla, Junta de Andalucía, Consejo Superior de Investigaciones Científicas (CSIC), 41092 Seville, Spain
5 Hospital Universitario Virgen Macarena, Servicio de Radioterapia, 41009 Seville, Spain; amadeoj.wals.sspa@juntadeandalucia.es
6 Hospital Universitario Virgen Macarena, Servicio de Medicina Nuclear, 41009 Seville, Spain; franciscoj.garcia.gomez.sspa@juntadeandalucia.es
* Correspondence: alplaza@us.es

**Abstract:** The purpose of this work is to present useful recommendations for the use of [18F]FDG-PET/CT imaging in radiotherapy planning and monitoring under different versions of EARL accreditation for harmonization of PET devices. A proof-of-concept experiment designed on an anthropomorphic phantom was carried out to establish the most suitable interpolation methods of the PET images in the different steps of the planning procedure. Based on PET/CT images obtained by using these optimal interpolations for the old EARL accreditation (EARL1) and for the new one (EARL2), the treatment plannings of representative actual clinical cases were calculated, and the clinical implications of the resulting differences were analyzed. As expected, EARL2 provided smaller volumes with higher resolution than EARL1. The increase in the size of the reconstructed volumes with EARL1 accreditation caused high doses in the organs at risk and in the regions adjacent to the target volumes. EARL2 accreditation allowed an improvement in the accuracy of the PET imaging precision, allowing more personalized radiotherapy. This work provides recommendations for those centers that intend to benefit from the new accreditation, EARL2, and can help build confidence of those that must continue working under the EARL1 accreditation.

**Keywords:** EANM guidelines; EARL accreditation; harmonization; FDG-PET; reconstruction protocol; radiotherapy planning; dose painting; FDG monitoring

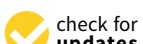



## 1. Introduction

A rising challenge in medicine is finding more accurate and personalized therapies. Advances in medical imaging are strongly linked to patient-tailored therapy planning, monitoring, and disease follow-up [1,2]. Since biological changes are expected during treatments, the functional information provided by nuclear medicine images plays an important role in clinical areas, such as oncology [3]. In the case of radiotherapy (RT), in which the prescribed dose is scheduled for multiple sessions, in addition to the diagnosis and staging of cancer, molecular imaging is also required to define the structures to be used in the planning process, which makes quantification and metrics a relevant process [4]. Significant advances in RT have allowed us to obtain an excellent balance between delivering a high dose to the tumor and a low dose to the healthy tissues surrounding the lesion [5,6]. Custom shielding blocks have been replaced with a versatile motorized

multileaf collimator (MLC) that allows computer-controlled linear accelerators (LINACs) to be connected to the treatment planning system (TPS), in which the dose distribution is calculated beforehand based on the image of the patient. In most cases, treatment planning is solved by considering the structures defined in the patient image from the computed tomography (CT) study. The role of image support in the whole process is so important that if some change is observed throughout treatment with image-guided radiotherapy (IGRT) techniques, adaptive planning must be considered [7,8]. In this scenario, where the highest precision in RT is dependent on the image, the provided information must be as complete as possible about the lesion for treatment. In this way, molecular images must be fused with morphological information from CT or magnetic resonance imaging data to be implemented in TPS for dose planning [9]. Positron emission tomography (PET) images can provide the visualization and quantification of the effects of treatment under monitoring to adapt the RT planning and dose prescription to the new targets if the procedure is ready to accurately consider biological changes [10].

The dose-painting technique (DP) is a new approach in RT where the prescription to the target volume is a non-uniform dose distribution based on functional information [11,12], usually provided by a PET study. Unfortunately, the quantification variability inherent to molecular imaging, such as that provided by positron emission tomography (PET) with [18F]fluorodeoxyglucose ([18F]FDG), is not ready for direct use in RT treatment planning. Beyond visual evaluation, the definition of the therapeutic target and the prescription dose based on parameters such as the standardized uptake value (SUV) require numerical values and standardization of imaging procedures [2,13]. In this scenario, the EARL [18F]FDG-PET/CT accreditation program ("resEARch for Life"-EARL) (EARL1) launched by the European Association of Nuclear Medicine (EANM) turned out to be essential for betting on personalized RT based on molecular imaging with guarantees. From then on, an increasing number of RT departments in many hospitals are being prompted to use the PET/CT image, as nuclear medicine departments are adopting the required guidelines and specifications to obtain the accreditation of their PET/CT devices and to be able to participate in multicenter studies. Today, specialists in nuclear medicine and radiation oncology work together to write procedure guidelines in which this accreditation program is specifically recommended to be followed for tumor imaging [14–16].

Unfortunately, but not unexpectedly, a normalization procedure shared by multiple PET/CT systems can cause the range of requirements to underuse some of the performance available in the latest generation of devices. This involves a loss of precision in spatial resolution for the target definition process, which could be essential in RT planning [17]. The standard in PET imaging for RT based on functional information should be based on the highest performance of the PET/CT devices [18], although harmonization conditions are required for multicenter studies. Kitajima et al. [19] stated how necessary harmonized quantitative volume-based values obtained with [18F]FDG-PET are to provide essential information regarding prognosis for both recurrence and death in patients with operable invasive breast cancer. Ly et al. [20] observed significant differences in a recurrent parameter to classify the status of the disease and stratify patients with lymphoma, when the updated EARL recommendations were used compared with previous criteria. Some studies propose SUV harmonization strategies that can improve the detection of lesions by using reconstruction algorithms that comply with older systems to allow comparison with historical case cohorts [21]. Other solutions are based on the use of a specific software tool, such as Siemens's EQ.PET, capable of achieving both goals from a single data set with excellent results [22], although this software can be applied only to scanners and reconstruction algorithms developed by the same company and also works as a black box without the ability to check the image segmentation result by the specialist.

In this scenario, since the implementation of EARL1, it has been proposed to update this accreditation to a new standard (EARL2) [23] for the necessary harmonization process [24,25]. Although this new accreditation is recent, studies have already been carried out to evaluate the impact on clinical practice of the implementation of EARL2 [18,20]. Despite

the previous works presented as comparison studies to validate software tools such as EQ.PET and assess the ability of several approaches to harmonize SUVs from different PET systems by using multiple reconstruction techniques, it would also be interesting to know in which clinical situations it is convenient to make use of one or the other accreditation, without sacrificing any of them.

The significant impact of PET imaging parameters on automatic tumor delineation for RT planning has been extensively shown. In this work, we tried to establish useful recommendations for the use of PET/CT imaging in RT planning and monitoring [26]. We performed a comparative study between the planning of actual cases in which the volume segmentation process was carried out based on image data obtained with different accreditation protocols. To simplify a rather complex and multivariable problem, a proof-of-concept experiment based on a specific phantom was carried out. This previous experiment provided us with the relevant characteristics that the lesions must present in order to cover a wide spectrum of situations with as few cases as necessary for this comparative study.

## 2. Materials and Methods

### 2.1. New EARL Accreditation

The EARL [18F]PET/CT accreditation consists of two procedures performed with phantoms. First, the 6 L cylinder with 70 MBq of [18F]FDG is scanned in two or more beds with a typical duration per bed (5 min). The uniformity of scan is checked, as well as the correspondence of the calculated SUV with the reference and expected value of 1. An error of ±10% is accepted. This procedure verifies the well counter measurements for dose preparation with the accuracy of the SUV provided by the reconstruction software and the internal calibration with the daily quality control of PET/CT system. This procedure is the same for the EARL1 and EARL2 standards. The second procedure consists of the NEMA-2007 phantom, which simulates six tumors over the background. The 10 L phantom is filled with 20 MBq of [18F]FDG. The six glass spheres with diameters from 10 to 37 mm are filled with [18F]FDG at ten times the concentration of the background. The EARL1 standard checks the recovery coefficient (RC) using SUVmax and SUVmean on the volumes of interest delineated on the spheres. RC is the ratio between the concentration of the measured and the actual activity of each spherical volume. The EARL1 standard accepts RC in the range 0.27–0.43, EARL2 in the range 0.39–0.61, allowing and enforcing higher quality and correspondence with the activity concentration in the tumor. EARL2 introduced SUVpeak, which averages uptake in a 12 mm diameter VOI positioned such as to yield the highest value across all tumor voxels. SUVpeak replaced SUVmax, which is prone to noise, and SUVmean, which is affected by manual delineation of the tumor. Higher RC values in EARL2 require generally higher spatial resolution of the PET scanner and working with images with smaller pixel size.

For this work, the PET/CT scanner used was the Siemens Biograph mCT 64 model. This device achieved EARL1 accreditation in 2017 [27]. The image reconstruction protocol that met accreditation criteria did not achieve the best resolution that the equipment could provide, underexploiting its capabilities. Therefore, in 2019, to achieve the best performance of this device, new parameters were followed, and recently, we renovated the annual EARL accreditation, which was called EARL2 [23].

The analysis of the images obtained for both studies was carried out with IDL Virtual Machine software [28] and required the calculation of RCs. In EARL1, these RCs were calculated from different values: SUVmean, which is the result of dividing the average SUV inside the volume by the total volume of the sphere, and SUVmax, which is obtained by dividing the maximum SUV inside the volume by the total volume. The new EARL2 accreditation also uses SUVpeak, which considers a spherical volume of 12 mm in diameter over the original volume to define the SUV corresponding to the highest uptake.

Previously, for the reconstruction of the EARL1 image, the iterative algorithm "ordered subset expectation maximization" (OSEM) was used, as well as a post-processing Gaussian filter with full width at half maximum (FWHM) equal to 6 mm. In the case of EARL2, the

reconstruction algorithm applied was TrueX, based on the point spread function (PSF), and a Gaussian filter with FWHM = 5 mm. The time-of-flight (TOF) correction was applied for both accreditations. For EARL1, the PET image grid size was $3.1819 \times 3.1819 \times 5$ mm$^3$, and $1.5910 \times 1.5910 \times 1.5$ mm$^3$ for EARL2.

Reconstructions of the anthropomorphic phantom PET/CT image fit the requirements of EARL2 with several parameters, which were established when the RCs appeared to be more centered between the limiting values established by the accreditation and so to be as close as possible to other accredited devices. The RCs obtained for EARL1 and EARL2 are shown in Figure 1. As expected, while in the EARL2 reconstruction most volumes showed very similar RCs (except the smallest), in the EARL1 reconstruction, these RCs showed greater differences between them. As will be discussed later, these results could have some clinical influence when EARL1 reconstructions are used in clinical cases with lesions of various sizes.

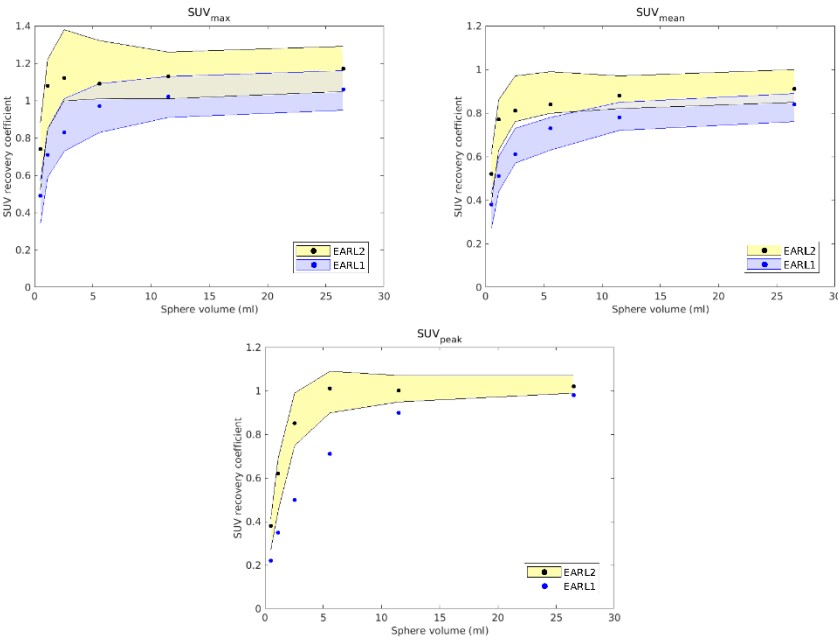

**Figure 1.** SUV recovery coefficients for EARL1 (blue dots) and EARL2 (black dots) against the volume of each sphere for each chosen reconstruction. The lines with the accepted limits for EARL1 appear in blue and yellow for EARL2. SUVpeak is introduced in EARL2 and did not exist in EARL1.

## 2.2. Resampling Process

To calculate the dose in the clinical image, different interpolations in the PET and CT images are necessary. First, it is necessary to interpolate the size of the PET image with the CT grid size for the segmentation of the volumes of interest (VOI) in the resulting fused image. In the second step, the image must be interpolated to the dose calculation grid, which is selected as a compromise with the accuracy involved in each case, depending on the VOI sizes and expected dose gradient, fundamentally. For this work, we established the size of $256 \times 256$ pixels per axial slice. This resolution is higher than that usually considered in commercial treatment planning systems despite the higher computational time involved, but our planning system was developed to run on a multiprocessor platform to achieve solutions in a reasonable computational time while maintaining high precision, which was necessary for this work to make a fair comparison between the treatment plans.

For this work, a configuration of an experiment with an anthropomorphic phantom (CIRS 606 model) hosting known different volumes and SUV distributions was used to find the interpolation method that best provided actual volumes [10]. Two inserts with different volumes and filled with a radioactive solution of [$^{18}$F]FDG were used to simulate two tumors of different sizes. The first insert consisted of an Eppendorf tube of known

volume V1 (0.3 mL, with an activity of 0.116 MBq of [$^{18}$F]FDG), and another identical tube called V2 (0.3 mL) was placed inside a cryovial tube V3 (2 mL, with an activity of 0.1 MBq). This configuration presented a scenario as generic as possible, to take into account different lesions ranging from very small volumes to larger ones, and with both homogeneous and heterogeneous activities. In this way, this proof-of-concept experiment allowed us to assess beforehand the relevant scenarios to be analyzed in order to reduce the actual cases needed for the comparison study.

The image acquisition of this set-up was carried out using both EARL protocols, and three typical 3D interpolation methods were studied: linear, nearest neighbors, and spline. Segmentation of the volumes of uptake was performed in each grid using an algorithm based on affine propagation [29]. The parameters associated with this algorithm were modified to generate all volumes to simultaneously achieve the set of values closest to the actual corresponding volumes. Once established, these parameters were kept constant in all subsequent reconstructions and interpolations. In the interpolation process from the PET grid to the CT grid, and from this CT grid to the dose calculation grid, a comparative analysis of the volumes generated by segmentation with the known volumes was performed to find the best method to represent the potential actual volumes in patients. In addition, the similarity in the shape and relative position of the volumes was evaluated. For this, the shape coefficient (SC) of each segmented volume was considered as the division between the intersection and the union of the obtained and actual volumes. Therefore, in the ideal case, SC = 1.

### 2.3. Volume Segmentation Method and Radiotherapy Planning

Intensity-modulated radiation therapy (IMRT) plans were calculated for all cases from the segmentation and generation of planning target volumes (PTVs) and organs at risk (OARs). The whole planning process was carried out on the CARMEN platform [30], where image processing and segmentation were performed, and an accurate dose calculation was considered by MC simulation. For this work, a forward planning algorithm, such as BIOMAP [30] implemented on the CARMEN platform, was required, where a direct aperture of the MLC is based exclusively on segmented morphofunctional volumes in patient images [10]. Any change in volumes inherent to the followed accreditation along with the segmentation process had some distinguishable influence on the optimization process.

Therefore, the MLC apertures for the radiotherapy plans were generated considering only the data from the PET/CT images, regardless of the desired dose. The clinical impact on the treatments due to the different reconstructions of the PET image could be directly assessed on the dose calculated through this type of optimization. Treatment planning was conducted independently for the volumes obtained from both reconstructions.

The general procedure for tumor imaging acquisition was taken from EANM guidelines [24]. Since the main objective of our study was to observe the greater clinical impact associated with the establishment of one accreditation protocol or the other, certain conditions were forced into a routine clinical scenario, such as a stress test for both accreditations. In this sense, the differences found could be considered the largest expected in any scenario. Semi-automatic segmentation of metabolic active tumor volumes (MATVs) is recommended for each case by setting a threshold of 41% of the SUV maximum in the region of interest around each lesion. We adopted this single value for all lesions in each evaluated case, since the clinical routine could lead to not repeating the same process on the same image, which is usual in RT when the prescription is based exclusively on the morphological CT image. However, all cases presented higher tumor-to-background values and homogeneous backgrounds to consider 41% of the maximum SUV as defined in the EANM guidelines [31]. Subsequently, an affine propagation-based segmentation algorithm [29] was applied to differentiate heterogeneous areas within the same lesion. Next, the transition from the CT grid size (512 × 512 voxels per slice) to the dose calculation grid (256 × 256 voxels per slice) was carried out using the interpolation methods chosen after the previous study with the anthropomorphic phantom.

## *2.4. Clinical Cases*

The study was performed in accordance with the ethical standards of our institutional research committee and with the Declaration of Helsinki of 1964 and its later amendments. Informed consent was obtained from all patients involved in the study.

All patients prior to [$^{18}$F]FDG-PET fasted overnight. The prescribed dose was in a range of 3–4 MBq/kg (resulting in 230 MBq per 70 kg patient). The patients were weighed prior to the study. CT was performed as necessary for respective EARL accreditation. For EARL1, the CT image had a size of 256 × 256 pixels at an 80 × 80 cm field of view with an axial slice thickness of 5 mm, and for EARL2, a size of 512 × 512 pixels at the same field of view and axial slice thickness. PET images were acquired following the protocols described above for EARL1 and EARL2 accreditations.

The clinical cases presented below were selected according to the differences between them regarding morphology, size, location, and number of lesions with the idea of covering a wide range of possible scenarios for the DP approach. In fact, the range of prescriptions for DP was previously considered in the study conducted with the anthropomorphic phantom [10]. Just like in the phantom, the different sizes considered in all clinical cases were relatively small, since, as it appears in Figure 1, no potential differences between accreditation protocols were expected for volumes larger than 5 mL.

### 2.4.1. Case 1

A case of head-and-neck cancer was evaluated with a single morphological lesion but with two regions in the PET image for different dose prescriptions. This was a cervical paraganglioma located next to the right parotid gland where two different size targets were considered according to the heterogeneity of [$^{18}$F]FDG uptake in the PET study with lesions (Figure 2). The OARs considered in treatment planning were the spinal cord, the left and right parotid glands, the larynx, and the jaw. The planning objectives were that at least 90% of the internal target (PTV2) must receive 56 Gy, and at least 90% of the external target (PTV1) must receive 50 Gy.

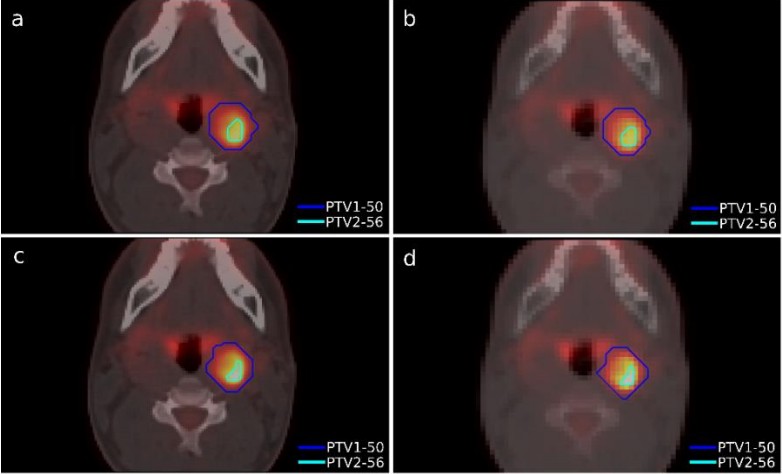

**Figure 2.** Segmented structures as targets in Case 1 using accreditations from EARL1 (top) and EARL2 (bottom). Contours in the isocenter axial slice of PTV1 and PTV2 in the PET/CT grid (**a**,**c**) and in the dose calculation grid (**b**,**d**). The color scale is the same for all figures.

### 2.4.2. Case 2

A case of lymphoma with a single lesion located in the right lung close to the sternum was selected (Figure 3). As in the previous case, the lesion presented some heterogeneity in [$^{18}$F]FDG uptake, so, although this case presented only one morphological volume, two different regions with different dose prescriptions were distinguished thanks to functional information from PET study. Additionally, this case involved uncertainty in breathing

movement for further discussion. The OARs considered in treatment planning were the left and right lung, the heart, the spinal cord, and the esophagus. The planning objectives were that at least 90% of the internal target (PTV2) must receive 36 Gy, and at least 90% of the external target (PTV1) must receive 30 Gy. Furthermore, a sub-volume was generated in the lung that contained the lesion to quantify the different doses delivered to the whole lung and the region closest to the tumor to assess the influence of breathing movement, since it could be an important aspect to consider when one or the other accreditation protocol should be chosen.

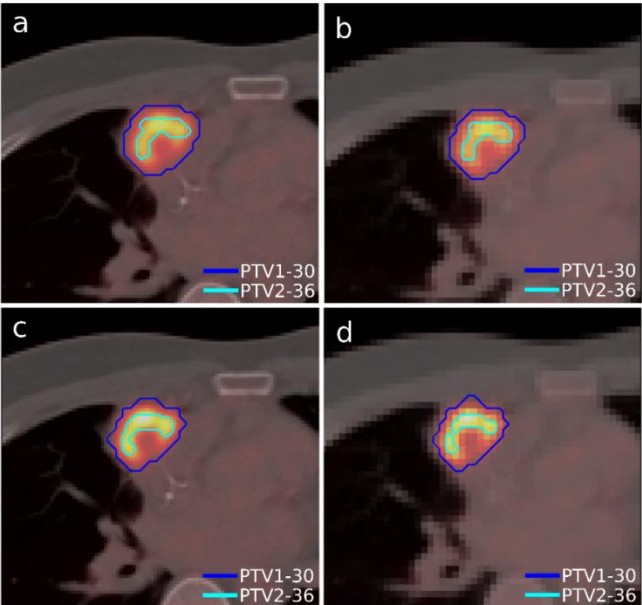

**Figure 3.** Segmented structures as targets in Case 2 using accreditations from EARL1 (top) and EARL2 (bottom). Contours in an isocenter axial slice of PTV1 and PTV2 on the PET/CT grid (**a**,**c**) and in the dose calculation grid (**b**,**d**). The color scale is the same for all figures.

### 2.4.3. Case 3

Another case of lung lymphoma was studied. Unlike the previous cases, there were two disconnected lesions of different sizes, one in the right lung and the other in the left (Figure 4). The OARs involved in this study were the same as in the previous case, and similarly, two auxiliary structures were generated in both lungs. For this case, the planning objectives were at least 90% of each target (PTV1 and PTV2) receiving 36 Gy.

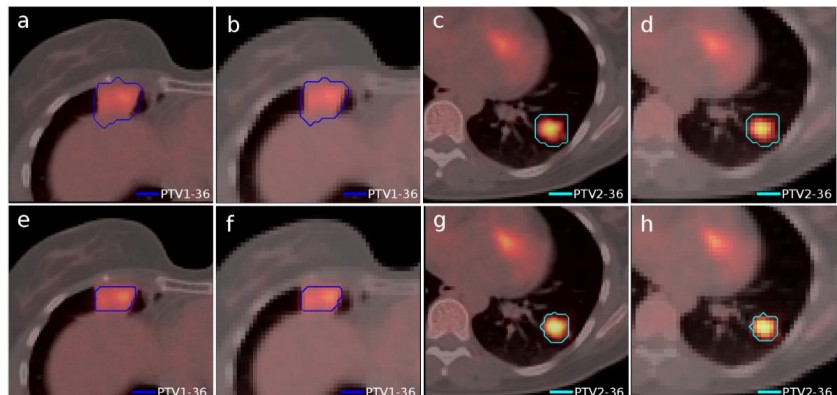

**Figure 4.** Segmented structures as targets in Case 3 using accreditations from EARL1 (top) and EARL2 (bottom). Contours in an isocenter axial slice of PTV1 and PTV2 in the PET/CT grid (**a**,**c**,**e**,**g**) and in the dose calculation grid (**b**,**d**,**f**,**h**). The color scale is the same for all figures.

## 3. Results

### 3.1. Segmentation of PET Images from the EARL1 and EARL2 Reconstructions

The results obtained for the study of the interpolation procedure for each reconstruction method are presented in Tables 1 and 2. The volume values obtained after the two involved interpolations (from PET grid to CT grid, and from CT grid to dose calculation grid) are shown for the three assessed 3D interpolation methods, i.e., linear, nearest neighbors (NN), and spline, by considering all the possible combinations over both interpolation processes. Tables 3 and 4 show the SC of the volumes V1, V2, and V3 hosted in the phantom obtained for the different interpolation methods applied to each of the reconstructions involved in the accreditations EARL1 and EARL2, respectively. To consider the overall effect of each interpolation method, the coefficients obtained for all volumes in Tables 3 and 4 were multiplied. Table 5 shows these cumulative coefficients for each interpolation method in both accreditations throughout the resampling process.

**Table 1.** Obtained volumes with each evaluated interpolation method for EARL1 reconstruction. The deviation from the actual volume is shown in brackets.

| EARL1 | | | | | |
|---|---|---|---|---|---|
| **PET Grid–CT Grid** | **CT Grid–Dose Calculation Grid** | | | | |
| **IM** | IM | V1 (mL) | V2 (mL) | V3 (mL) | V2 + V3 (mL) |
| | L | 0.4874 (+62%) | 0.3621 (+21%) | 4.0662 (+103%) | 4.3029 (+87%) |
| **L** | NN | 0.4595 (+53%) | 0.3621 (+21%) | 3.5231 (+76%) | 3.8852 (+69%) |
| | S | 0.5292 (+76%) | 0.4038 (+35%) | 4.1637 (+108%) | 4.4004 (+91%) |
| | L | 0.6406 (+114%) | 0.5709 (+90%) | 3.6902 (+85%) | 4.0662 (+77%) |
| **NN** | NN | 0.4178 (+39%) | 0.3760 (+25%) | 2.8129 (+41%) | 3.1889 (+39%) |
| | S | 0.6406 (+114%) | 0.5709 (+90%) | 3.6902 (+85%) | 4.0662 (+77%) |
| | L | 0.3899 (+30%) | 0.5013 (+67%) | 3.5928 (+80%) | 3.9130 (+70%) |
| **S** | NN | 0.3760 (+25%) | 0.3760 (+25%) | 3.0357 (+52%) | 3.4117 (+48%) |
| | S | 0.4595 (+53%) | 0.5152 (+72%) | 3.6485 (+82%) | 3.9687 (+73%) |

IM, interpolation method; L, linear; NN, nearest neighbors; S, spline.

**Table 2.** Obtained volumes with each interpolation method for EARL2 reconstruction. The deviation from the actual volume is shown in brackets.

| EARL2 | | | | | |
|---|---|---|---|---|---|
| **PET Grid–CT Grid** | **CT Grid–Dose Calculation Grid** | | | | |
| **IM** | IM | V1 (mL) | V2 (mL) | V3 (mL) | V2 + V3 (mL) |
| | L | 0.4038 (+35%) | 0.3621 (+21%) | 2.2838 (+14%) | 2.4787 (+8%) |
| **L** | NN | 0.3481 (+16%) | 0.3064 (+2%) | 1.9913 (−0.4%) | 2.2977 (−0.1%) |
| | S | 0.4595 (+53%) | 0.3899 (+30%) | 2.3395 (+17%) | 2.5205 (+10%) |
| | L | 0.3621 (+21%) | 0.3203 (+7%) | 2.2977 (+15%) | 2.4648 (+7%) |
| **NN** | NN | 0.2924 (+3%) | 0.2924 (+3%) | 1.9496 (+3%) | 2.2420 (+3%) |
| | S | 0.4317 (+44%) | 0.3621 (+21%) | 2.3116 (+16%) | 2.4787 (+8%) |
| | L | 0.4456 (+49%) | 0.2924 (−3%) | 2.5066 (+25%) | 2.6737 (+16%) |
| **S** | NN | 0.3760 (+25%) | 0.2507 (−16%) | 2.2281 (+11%) | 2.4787 (+8%) |
| | S | 0.4874 (+62%) | 0.3064 (+2%) | 2.5205 (+26%) | 2.6876 (+17%) |

IM, interpolation method; L, linear; NN, nearest neighbors; S, spline.

**Table 3.** Shape coefficients obtained for the EARL1 reconstruction when going from the CT grid to the dose calculation grid.

| | | EARL1 | | | |
|---|---|---|---|---|---|
| **PET Grid–CT Grid** | | **CT Grid–Dose Calculation Grid** | | | |
| **IM** | IM | $SC_1$ | $SC_2$ | $SC_3$ | $SC_{2+3}$ |
| | L-S | 0.9211 | 0.8966 | 0.9766 | 0.9778 |
| **L** | L-NN | 0.7000 | 0.7931 | 0.7638 | 0.8148 |
| | S-NN | 0.6512 | 0.7742 | 0.7636 | 0.8140 |
| | L-S | 1.0000 | 1.0000 | 1.0000 | 1.0000 |
| **NN** | L-NN | 0.6522 | 0.6585 | 0.7623 | 0.7842 |
| | S-NN | 0.6522 | 0.6585 | 0.7623 | 0.7842 |
| | L-S | 0.8485 | 0.9730 | 0.9847 | 0.9860 |
| **S** | L-NN | 0.6667 | 0.7500 | 0.7565 | 0.8076 |
| | S-NN | 0.7143 | 0.7297 | 0.7455 | 0.7966 |

IM, interpolation method; L, linear; NN, nearest neighbors; S, spline; SCi, shape coefficient of volume i.

**Table 4.** Shape coefficients obtained for the EARL2 reconstruction when going from the CT grid to the dose calculation grid.

| | | EARL2 | | | |
|---|---|---|---|---|---|
| **PET Grid–CT Grid** | | **CT Grid–Dose Calculation Grid** | | | |
| **IM** | IM | $SC_1$ | $SC_2$ | $SC_3$ | $SC_{2+3}$ |
| | L-S | 0.8788 | 0.9286 | 0.9762 | 0.9834 |
| **L** | L-NN | 0.6875 | 0.6552 | 0.7746 | 0.8441 |
| | S-NN | 0.7059 | 0.6667 | 0.7571 | 0.8307 |
| | L-S | 0.8387 | 0.8846 | 0.9940 | 0.9944 |
| **NN** | L-NN | 0.6786 | 0.5714 | 0.7733 | 0.8470 |
| | S-NN | 0.6250 | 0.5667 | 0.7688 | 0.8424 |
| | L-S | 0.9143 | 0.9545 | 0.9945 | 0.9948 |
| **S** | L-NN | 0.6857 | 0.6957 | 0.7989 | 0.8404 |
| | S-NN | 0.7222 | 0.6667 | 0.7947 | 0.8366 |

IM, interpolation method; L, linear; NN, nearest neighbors; S, spline; SCi, shape coefficient of volume i.

**Table 5.** Shape coefficients obtained after the two consecutive interpolation processes (from the PET grid to the CT grid, and from the CT grid to the dose calculation grid) for each image reconstruction followed in EARL1 and EARL2 accreditations.

| **PET Grid–CT Grid** | **L** | | | **NN** | | | **S** | | |
|---|---|---|---|---|---|---|---|---|---|
| **CT Grid–Dose Calculation Grid** | **L** | **NN** | **S** | **L** | **NN** | **S** | **L** | **NN** | **S** |
| **EARL1** | 0,2725 | 0,1083 | 0,2471 | 0,2567 | 0,0659 | 0,2567 | 0,2449 | 0,0946 | 0,2481 |
| **EARL2** | 0,2307 | 0,0872 | 0,2319 | 0,1862 | 0,0583 | 0,1682 | 0,2765 | 0,1025 | 0,2764 |

L, linear; NN, nearest neighbors; S, spline.

*3.2. Planning of Clinical Cases under EARL1 and EARL2 Accreditations*

An assessment of each volume of interest was carried out for both reconstructions following the optimal interpolation methods for each stage, as was presented in the previous section. The corresponding volume quantifications are presented in Table 6.

**Table 6.** Segmented structures for each clinical case evaluated, prescription dose, and volume values in reconstructed images according to EARL1 and EARL2 accreditations.

| Clinical Case | Structure Name | Dose Prescription (Gy) | EARL1 Volume (cm$^3$) | EARL2 Volume (cm$^3$) |
|---|---|---|---|---|
| Case 1 | PTV1 | 50 | 16.6 | 15.5 |
| | PTV2 | 56 | 2.6 | 1.4 |
| Case 2 | PTV1 | 30 | 19.4 | 16.1 |
| | PTV2 | 36 | 3.8 | 2.7 |
| Case 3 | PTV1 | 36 | 41.5 | 19.8 |
| | PTV2 | 36 | 8.8 | 4.2 |

The dose distributions and dose–volume histograms of the planning solutions for the three cases are shown in Figures 5–10. In Figures 5, 7 and 9, the isodose lines of the three cases under evaluation were calculated for the volumes generated with EARL1 accreditation (left) and with EARL2 (right) on an axial slice of the EARL2 reconstruction, considering this as the protocol capable of generating structures more similar to actual lesions. The two targets corresponding to the EARL1 and EARL2 accreditations appear in blue and light blue regions, respectively. In Figures 6, 8 and 10, the dose–volume histograms of the calculated dose for EARL1 (dashed lines) and EARL2 (solid lines) solutions are shown for the volumes corresponding to EARL2, in the three cases.

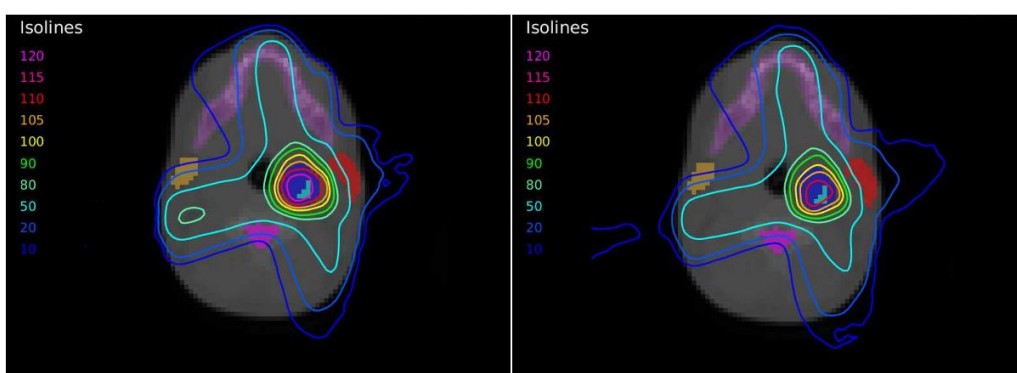

**Figure 5.** Dose distribution for Case 1 by applying the dose-painting technique under EARL1 (**left**) and EARL2 (**right**) accreditations.

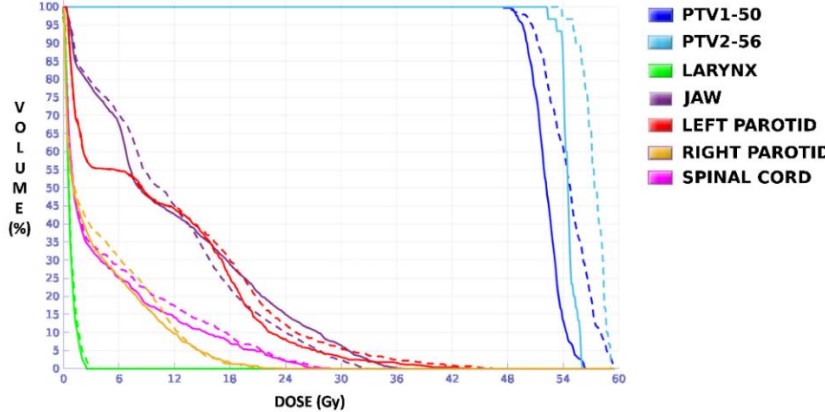

**Figure 6.** DVHs of the planning solutions for Case 1 under EARL1 (dashed lines) and EARL2 (solid lines) accreditations on the volumes generated with EARL2.

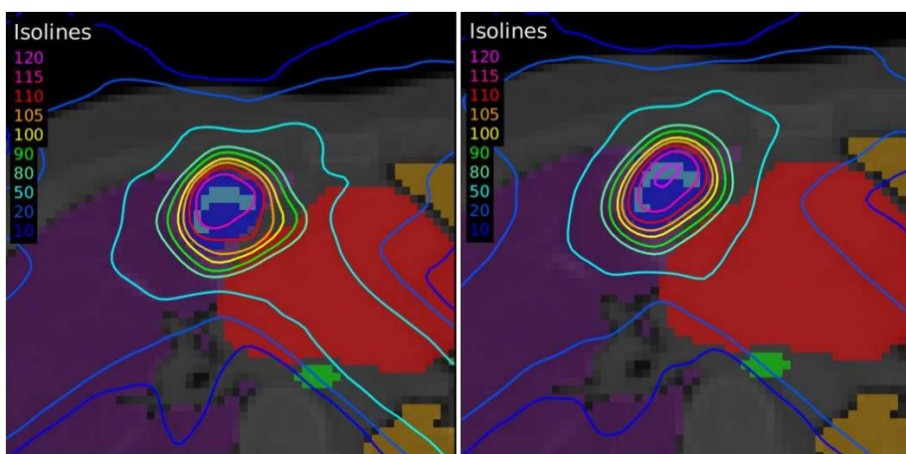

**Figure 7.** Dose distribution for Case 2 by applying the dose-painting technique under EARL1 (**left**) and EARL2 (**right**) accreditations.

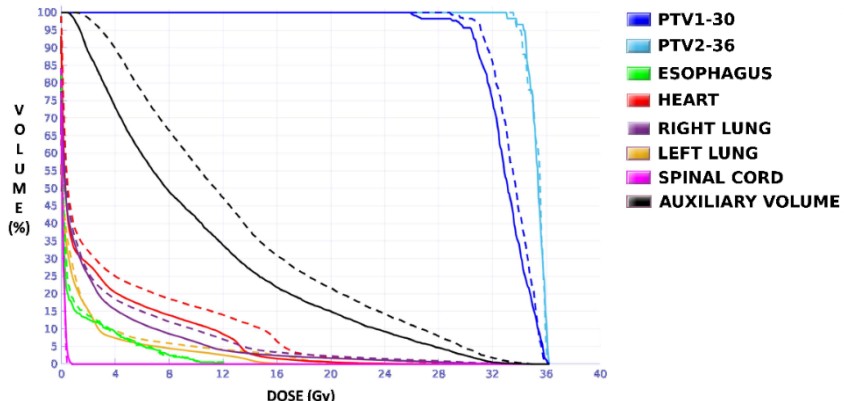

**Figure 8.** DVHs of the planning solutions for Case 2 under EARL1 (dashed lines) and EARL2 (solid lines) accreditations on the volumes generated with EARL2.

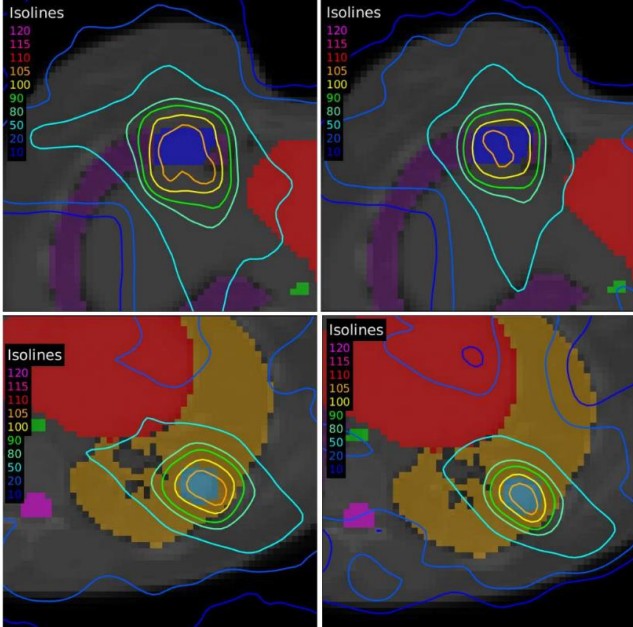

**Figure 9.** Dose distribution for Case 3 by applying the dose-painting technique under EARL1 (**left**) and EARL2 (**right**) accreditations.

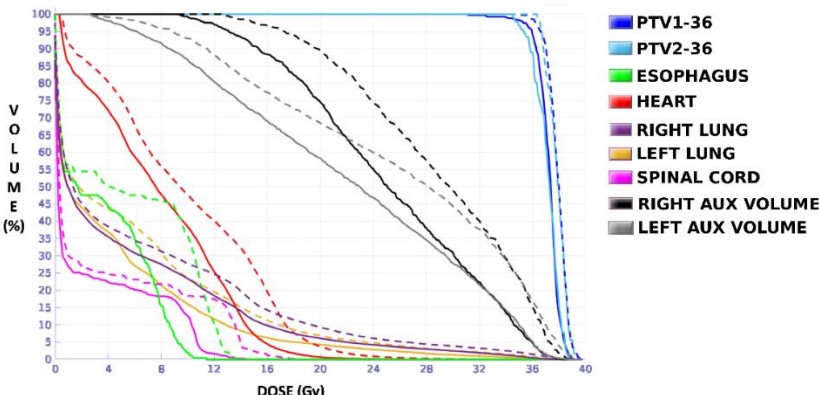

**Figure 10.** DVHs of the planning solutions for Case 3 under EARL1 (dashed lines) and EARL2 (dashed lines) accreditations on the volumes generated with EARL2.

In general, isodose lines showed a slightly better conformation for EARL2 and, therefore, significant differences were not present in the DVHs for targets in all cases, except for Case 1, as will be discussed later. More important differences were found for OARs, which are also discussed next.

## 4. Discussion

Although a limited number of cases were evaluated, sufficiently representative situations for assessing EARL accreditation impact were considered based on a proof-of-concept experiment designed on an anthropomorphic phantom. Otherwise, this study would have been difficult to carry out, since many factors are involved in the whole process of radiotherapy planning when PET/CT images are used for segmentation of target volumes.

In this proof-of-concept experiment, the volume deviations obtained (Tables 1 and 2) with respect to the actual known volumes V1, V2, and V3 hosted in the phantom showed that the EARL1 accreditation generated volumes deviating further from the actual values and always larger than those obtained with the EARL2 accreditation. The optimal interpolation procedure for each reconstruction was established as a compromise between the evaluation of the shape coefficients in Table 5 obtained after the interpolation process and the evaluation of the segmented volumes shown in Tables 1 and 2, obtained with each interpolation method. The methods chosen for EARL1 were spline for the 3D interpolation from the PET grid to the CT grid and linear from the CT grid to the dose calculation grid, and linear for EARL2 in both interpolation processes. These interpolation methods did not always show the best results considered individually, but their combination throughout the whole process provided more uniform values for the different volume sizes and an adequate and robust form factor. It is remarkable that, contrary to our expectations, the spline method was not the best selection. The diffuse character of the [$^{18}$F]FDG signal in the PET image does not sufficiently contribute to generating morphological irregularities to make the spline method more appropriate than others.

For the clinical cases, as expected, EARL1 always provided larger volumes than EARL2, and greater differences in size were found between both accreditations for smaller structures, consistent with the observations reported by Kaalep et al. [18]. Furthermore, cases involving lesions or targets of different sizes, where some of them were smaller than 5 mL, led to the worst result when the accreditation of EARL1 was followed. On the other hand, the increase in the volume of the structures observed for the EARL1 accreditation caused an increase in the level of undesirable doses in the OARs and in the adjacent regions. In the case of moving targets such as the lungs, this increasement may become more relevant, since the healthy tissue surrounds the target.

The plan calculated for the EARL1 accreditation and evaluated on the volumes generated with images from the EARL2 accreditation delivered a higher dose on the structures, as can be seen in Figures 6, 8 and 10. Planning for EARL1 covered the prescribed dose of

both PTV1 and PTV2 from EARL2 reconstruction because EARL1 volume sizes were larger. The toxicity limits for OARs were not compromised with this overdose region around PTVs under EARL1 accreditation.

Unlike in Case 1, the therapeutic volumes of Cases 2 and 3 were surrounded by large OARs, the lungs, so the high dose administered in the vicinity of the lesion could be clinically relevant due to the risk of damaging healthy tissue. For this reason, although the DVHs did not show important differences in the doses to the PTVs generated with both accreditations, an overall increase in the dose to the OARs was caused with EARL1 planning (Figures 8 and 10). It is important here to consider the inherent uncertainty associated with breathing in lung cases to evaluate the real clinical impact. For this reason, the auxiliary structures were generated for planning to evaluate both the movement of the lesion due to respiration and the undesirable dose in the region surrounding the PTVs. The greatest differences were observed in these auxiliary volumes. It can be concluded that with the EARL2 reconstruction, it is possible to achieve the corresponding dose prescription with greater spatial accuracy than with the EARL1 reconstruction, and, therefore, to avoid delivering an additional low dose to the lung around the lesion. The use of EARL1 accreditation could be considered acceptable in these cases, as this spatial uncertainty is within the uncertainty associated with respiratory movement. However, in those radiotherapy techniques, such as stereotactic ablative radiation therapy (SABR), where an important escalation of the prescription dose is planned, we would recommend considering the breathing movement of the lesion and identifying all phases of the respiratory cycle by a segmentation procedure based on the EARL2 accreditation.

In the EARL2 reconstruction, the RCs were always higher than the corresponding ones in the EARL1 reconstruction (Figure 1) and closer to 1. If these coefficients are less than 1, this implies that the maximum value of the SUV obtained after the reconstruction will be less than the value of the actual maximum SUV. This will cause a greater extension and diffusion and, therefore, an increase in volumes after reconstruction. On the contrary, in EARL2 accreditation, the RC values were higher than those in EARL1, and therefore, they generated smaller volumes, closely corresponding to the reconstructed structures. This is especially relevant for volumes less than 2.5 mL, as can be seen in Figure 1. Only Case 1 presented a PTV with this small size, and this case showed the most important differences in the DVHs of the corresponding PTVs for the two accreditations.

The differences between the EARL1 and EARL2 reconstruction settings were even more remarkable in Case 3 where multiple, unconnected lesions with different sizes and the same prescription were presented. Again, our results were consistent with observations of Kaalep et al. [18]. This could be related to the level of demand for the different voxel sizes in each accreditation. While in EARL1, the adjustment is approximately logarithmic, for EARL2 accreditation, this adjustment is practically linear, except for volumes smaller than 0.5 mL. Therefore, we could say that in the scenario of multiple lesions of different sizes for the same prescription dose, the EARL2 accreditation becomes more necessary to avoid a clinically relevant inaccuracy in the definition of target volumes from PET/CT images under EARL1 accreditation.

## 5. Conclusions

This work was aimed at answering the question of how necessary a new PET accreditation is when reconstructing therapeutic targets for radiotherapy planning. Generic analysis of sizes, shapes, and locations was established by using known volumes with [$^{18}$F]FDG inside a specific phantom to find out which interpolation method for image fusion and grids for dose calculation throughout the planning process was able to achieve the highest precision for defining different uptake volumes. This analysis concluded that the EARL2 accreditation generates smaller volumes and is more similar to the actual sizes than the EARL1 PET accreditation, providing an increase in accuracy that could be significant in the planning process and dose calculation, depending on the relative sizes of the targets in each case. To illustrate this, several clinical cases were chosen representing the most sensible

scenario due to the sizes of the therapeutic target and were evaluated following both EARL accreditations, in order to observe the clinical impact.

In summary, we think the new EARL accreditation represents an advance in the reconstruction of the PET image for its implementation in the treatment planning process and subsequent monitoring. The volumes generated maintain greater precision when defining tumor lesions and their physiological extension, resulting in a clear advantage over the previous EARL reconstruction. For this reason, this work supports the application and propagation of the new EARL accreditation to achieve better accuracy when performing a more personalized radiotherapy treatment and monitoring purposes are considered. However, it was verified that, under the previous EARL accreditation conditions, the targets that would be generated by the new EARL accreditation would receive the prescribed doses. Therefore, this work also helps reduce concerns in those centers that, having somewhat less advanced devices, need to continue working under previous accreditations and do not plan to implement ambitious techniques, such as SABR or monitoring cases with breathing move.

**Author Contributions:** Conceptualization, E.J.-O. and A.L.; methodology, E.J.-O. and A.L.; software, E.J.-O., R.A. and A.U.; validation, E.J.-O. and A.L.; formal analysis, E.J.-O. and A.L.; investigation, E.J.-O., R.A. and A.L.; resources, E.J.-O., R.A., M.B., A.W.-Z., F.J.G.-G. and A.L.; data curation, E.J.-O., R.A. and A.L.; writing—original draft preparation, E.J.-O. and A.L.; writing—review and editing, E.J.-O., A.U., M.B. and A.L.; visualization, E.J.-O., R.A. and A.L.; supervision, A.L.; project administration, A.L.; funding acquisition, A.L. All authors have read and agreed to the published version of the manuscript.

**Funding:** Project P20_01053 funded by the European Union and the Junta de Andalucía through the European Regional Development Fund (FEDER).

**Institutional Review Board Statement:** All procedures performed in studies involving human participants were in accordance with the ethical standards of the institutional and/or national research committee and with the Declaration of Helsinki of 1964 and its later amendments or comparable ethical standards. This study was approved by the Ethics Committee of the Hospital Universitario Virgen Macarena de Sevilla for the study entitled: "Integración de la imagen PET/CT en una aplicación radioterápica de precisión y adaptative" with reference C.P. DP-PET/CT-C.I. 1958-N-20.

**Informed Consent Statement:** Informed consent was obtained from all subjects involved in the study.

**Data Availability Statement:** The data presented in this study are available within the article. Further data are available on request from the corresponding author.

**Conflicts of Interest:** The authors declare no conflict of interest. The funders had no role in the design of the study; in the collection, analyses, or interpretation of data; in the writing of the manuscript, or in the decision to publish the results.

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
