# Peer review of "Implications of the Harmonization of [18F]FDG-PET/CT Imaging for Response Assessment of Treatment in Radiotherapy Planning"

_tomography, doi:10.3390/tomography8020090_

Round 1

Reviewer 1 Report

The manuscript entitled “Implications of harmonization 18F-FDG-PET/CT imaging for re- 2 sponse assessment of treatment in radiotherapy planningis an interesting case report, The results presented in this study are interesting and the manuscript was well written and organized. I suggest that this manuscript be considered for publication if minor revision is satisfied.

Following are the concerns need to be addressed

  1. The case reports describes about harmonization on 18F FDG PET-CT that assess the treatment in radiotherapy planning. Are there any previous reports of similar case of harmonization? If yes whats the age group reported
  2. Was there any prime feature of harmonization observed that can be reported particularly in this case?
  3. More recent literature should be cited, especially for review articles.
  4. What was the age group of the subjects that were enrolled?
  5. Interpretation of Figs. 5,6,7 and 9 should be expanded in the main text as these data are functional assay data and were generated to verify the predicted data
  6. Authors should give a better explanation for the sentence in line 355 “Unlike case 1, the MATV of cases 2 and 3 was inside large OARs, the lungs, so the 355 high dose administered in the vicinity of the lesion could be clinically relevant”
  7. Authors should expand the explanation on EARL1/2 accreditation in both introduction and conclusion section
  8. Authors should mention about the importance of experimental research on the present case report on harmonization for better knowledge also mentioning about the therapeutical approaches.
  9. Grammar errors should be fixed in several places

Author Response

Please see the attachment.
Thank you very much for your suggestions and comments. We have tried to answer all your questions and we honestly believe that the manuscript has improved substantially.

Reviewer 2 Report

The manuscript describes the advantages of a new EARL accreditation compared with the previous PET accreditation with 18F-FEG PET/CT in radiotherapy planning. The authors concluded that EARL1 accreditation generated much higher volumes and deviations than the proposed EARL2 method. The EARL2 strategy showed higher accuracy in defining the target volumes from PET/CT, enabling better dose calculation, treatment planning, and monitoring in clinical tumor cases.

However, this manuscript has several significant limitations:

1) only three cases were presented in the study. Therefore, it’s hard to provide statistical analysis to demonstrate the new accreditation is superior to a broad clinical case with greater precision, or whether the method can apply to more types of tumor lesions.

2) It’s unclear how and when the EARL2 method can provide reliable recommendations for clinical centers since EANM has launched EARL1 with guarantees.

3) the manuscript needs significant revisions in grammar, such as “…observed, what produced higher doses in the organs”, “…for RT planning has been extensively shown”, “…provided us with the relevant characteristics that the lesions had to present for…”, “DVH evaluation showed no important differences in the doses…”. Too many complex sentences with “in which”, “where” are used but not easy to follow.

4) many technical terms are either not defined, such as EARL, CARMEN, BIOMAP, or defined but not in their first appearance, such as PET. 3) the reference format is inconsistent, such as 13, 23, 25. Therefore, I do not recommend the manuscript for publication in Tomography.

Author Response

(The authors gave the same response as above.)

Reviewer 3 Report

In the paper entitled “Implications of harmonization 18F-FDG-PET/CT imaging for response assessment of treatment in radiotherapy planning” the authors describe a study aimed to evaluate clinical impact of the use of images fulfilling two different EARL accreditation strategies for the harmonization of PET devices. The motivation of the work seems very interesting and important for the clinical application of such harmonization strategies for PET imaging in the specific context of Radiotherapy planning. This manuscript represents a mixed experiment (with a preliminary preclinical component and a clinical component) that seems interesting and appear to deserve to be shared with medical and scientific community within this field. However, there are a couple of issues that need to be solved before its final acceptance. Globally, English writing should be reviewed by an expert and/or native speaker. Also, introduction misses some important points, while methodology is confusing and without reference to some important details. In such context, I would like to suggest the acceptance of this paper after a minor review.

Specific comments:

(Title) “Implications of harmonization 18F-FDG-PET/CT imaging for…” do not sounds and flows as expected. I would like to suggest: “Implications of the harmonization of 18F-FDG-PET/CT imaging for…”;

(Abstract) The proposed abstract does not give the appropriate information about theoretical background of the study and methodology used. It should be revised and include a short description of such topics;

(line 64) EARL accreditation programs should be better contextualized in what concerns to its aims and methods implicated;

(line 70) PET/CT prices?

(lines 74-76) “This involved loss of precision in spatial resolution for the target definition process could be essential [13] in the RT planning and the standard should be based on the highest performance” – too long and confusing. Sentence should be rephrased and ideas better explained;

(Material and Methods) Globally it seems that this chapter is written as a mixture of the description of methodological aspects together with discussion of results obtained. Attention should be paid to the global review of such point in the way to improve its scientific quality and soundness. Also, it should be clearly divided in methodological aspects related with the preliminary preclinical study with the use of phantom and in the study of clinical impact with patient cases;

(line 110-111) Sentence needs to be revised;

(line 160 and 162) nomenclature for the radiopharmaceutical not written in appropriate terms (special attention to the notation used for the radionuclide);

(line 221) a reference (11) is inserted but it appears to be a lapse or an error;

(clinical cases) Methodological information (protocol for image acquisition including radiopharmaceutical administered activity) is not provided. Ethical aspects not described until the end of the manuscript (with a single mention before references);

(Table 1-6) Tables are presented without any text interspaced. It could be revised in the way to help in the flow of the reading;

(Discussion) Literature published on EARL2 evaluations was not properly explored;

(Conclusions) This section seems to be too long, including aspects that should be present in the Discussion and do not helps with the flow of reading;

Author Response

(The authors gave the same response as above.)

Round 2

Reviewer 2 Report

The authors made significant revisions to the manuscript in the method section and data presentation. The authors justified that the current study is a proof of concept, providing guidelines for the comparative analysis with all the variances isolated, while the sample size is still considered small. Overall, I recommended the manuscript for publication in Tomography.

Author Response

Thank you very much for your reconsideration. Some changes related to grammatical corrections and style have been made. Without a doubt, you comments were essential to improve the manuscript.